# Management and Treatment Outcomes of High-Risk Corneal Transplantations

**DOI:** 10.3390/jcm11195511

**Published:** 2022-09-20

**Authors:** Karolina Urbańska, Marcin Woźniak, Piotr Więsyk, Natalia Konarska, Weronika Bartos, Mateusz Biszewski, Michał Bielak, Tomasz Chorągiewicz, Robert Rejdak

**Affiliations:** Chair and Department of General and Pediatric Ophthalmology, Medical University of Lublin, 20-079 Lublin, Poland

**Keywords:** cornea, corneal transplantation, high-risk corneal transplantation, keratoplasty, immunosuppression in corneal transplantation, corneal immune privilege

## Abstract

Corneal transplantation is the most effective treatment for corneal blindness. Standard planned keratoplasties have a high success rate. Conditions such as active inflammation at the time of surgery, the presence of ocular surface disease, previous graft disease, or neovascularization make them more susceptible to rejection. These are so-called high-risk corneal transplantations. In our study, we selected 52 patients with a higher risk of graft rejection. A total of 78 procedures were performed. The main indications for the first keratoplasty were infections (59.6%) and traumas (21.2%). Visual acuity (VA) significantly improved from 2.05 logMAR on the day of keratoplasty to 1.66 logMAR in the latest examination (*p* = 0.003). An analysis of the graft survival showed a 1-year survival of 54% and a 5-year survival of 19.8% of grafts. The mean observation time without complications after the first, second, and third surgery was 23, 13, and 14 months, respectively. The best results were noted among patients with infectious indications for keratoplasty (*p* = 0.001). Among them, those with bacterial infection had the best visual outcomes (*p* = 0.047).

## 1. Introduction

Corneal disorders are the third leading cause of blindness in the human population, after cataracts and glaucoma [1]. They can be managed very effectively with keratoplasty, during which the damaged cornea is replaced with healthy tissue. Corneal grafting is the most common type of transplantation performed worldwide [1]. It is also one of the most successful ones, as it carries a low risk of graft rejection due to corneal avascularity [2]. However, in some cases, there is a higher risk of corneal transplant failure. It may be due to different factors leading to the loss of corneal immune privilege, such as ocular surface diseases or active inflammation at the time of surgery. In the case of infections unresponsive to conservative treatment or ocular traumas, the integrity of the eyeball is endangered, and the keratoplasty has to be performed urgently. The inability to examine the structures of the eyeball due to corneal opacity is another indication of urgent keratoplasty. The risk of graft failure is also higher in patients with a prior clinical history of transplant rejection or other eye surgeries, particularly glaucoma surgery [3].

Corneal transplantation is the only possible treatment in the case of extensive corneal lesions. The surgical technique depends on the size, location, cause, and depth of the corneal damage. In recent years, there have been significant advances in the treatment modalities for corneal blindness. One of the greatest breakthroughs was the introduction of endothelial keratoplasty. It provides a significantly lower risk of transplant rejection, faster visual recovery, and longer transplant survival than the traditional penetrating keratoplasty. Endothelial keratoplasty procedures include Descemet’s automated endothelial keratoplasty (DSAEK) and Descemet’s membrane endothelial keratoplasty (DMEK) [4].

The aim of this study was to assess the functional and structural outcomes of high-risk corneal transplantations. Specifically, we focused on different indications for keratoplasty in terms of their impact on graft rejection risk and visual outcomes. We intended to point out the potential corneal blindness etiologies whose management should be reconsidered to improve the functional and anatomical success rates of keratoplasty. In addition, we compared our results with other high-risk corneal transplantation reports found in the literature.

## 2. Materials and Methods

This retrospective case series involves patients of the Department of General and Pediatric Ophthalmology at the Medical University of Lublin, Poland. The study followed the tenets of the Declaration of Helsinki and was based on the data of penetrating keratoplasties performed between 2018 and 2022. In the case of patients with a prior history of keratoplasty, the surgeries performed before 2018 were also investigated. The inclusion criterion was a higher risk of graft failure. We considered the so-called hot-grafting with ongoing active inflammation (due to corneal trauma, burn, perforation, or infection) and previous graft rejection to be a high-risk setting. Previous graft rejection was qualified as a primary high-risk transplant only when the indication for the first keratoplasty was not related to a higher risk of graft failure. Patients without risk factors or with insufficient data were excluded from our research. The data extracted from the medical records included: surgical technique, history of surgical interventions prior to keratoplasty, pre- and postoperative visual acuity (VA), graft diameter, and systemic steroid therapy or immunosuppressive therapy with Mycophenolate mofetil (MMF) following the transplantation. VA was evaluated in all patients using Snellen's original test with conversions to decimal and logMAR scales for statistical analyses. Lower VAs were classified as follows: counting fingers, hand motion, light perception, or no light perception. VAs were assigned with logMAR scores of 1.9, 2.3, 2.7, and 3.0, respectively [5,6]. Bacterial and fungal infections were confirmed by the isolation and identification of the pathogens in microbiological testing. Viral infections were confirmed based on clinical presentation and slit-lamp examination. Functional success was described as an improvement in VA from the baseline to the most recent follow-up. For each analysis found in the results section, we selected those patients who met the analyzed criteria and provided enough data to include them in the specific analysis.

### Statistical Analysis

Statistical analysis was performed using R programming language and RStudio: Integrated Development Environment for R language, software version number: 2022.7.1.554, author: RStudio Team (2022), Boston, MA, USA. All statistical tests were performed with 95% statistical significance. The Shapiro–Wilk test was used to examine the normality of distributions. The Chi-Squared Test of Independence was used to examine the difference in postoperative VA depending on primary etiology. Differences between pre- and postoperative VA depending on etiology and type of infection were examined with the Kruskal–Wallis test. The relationship between etiology and the time span from the first to the second keratoplasty was also examined with the Kruskal–Wallis test. The Wilcoxon test and Student’s *t*-test were used to examine the improvement in VA. The Kaplan–Meier estimator was used to examine the survival of the first grafts. In the uncomplicated cases, we included the time between keratoplasty and the last follow-up. In the complicated cases, we used (a) the time between the first and second keratoplasty, (b) the time between keratoplasty and the discovery of an atrophic eyeball in the postoperative follow-up, (c) the time between keratoplasty and the enucleation of the eyeball. The Mantel–Haenszel test was used to examine the differences in survival probability in different groups.

## 3. Results

Out of the 120 patients submitted to penetrating keratoplasty, we selected 52 patients with a total of 78 keratoplasty procedures performed. The group consisted of 30 men and 22 women with an average age of 58.7 ± 18.7 years (range 22–97 years). Further analysis revealed two peaks in patients’ age (Figure 1). The main indications for the first keratoplasty were infections (59.6%) and traumas (21.2%) (Figure 2). More information about the study group and indications for the first keratoplasty can be found in Table 1.

The mean observation time without complications after the first, second, and third surgery was 23, 13, and 14 months, respectively. Twenty-two patients (42.3%) required a second keratoplasty. In four cases (7.7%), a third keratoplasty was necessary. The average time between the first and second and between the second and third keratoplasty was 17 and 27 months, respectively. Thirty-two patients (61.5%) noted an improvement in VA. In nine cases (17.3%), VA remained unchanged, and in another nine cases (17.3%), it deteriorated. Two patients did not provide information about their preoperative and postoperative VA. An analysis of the average values of VA converted to logMAR indicated an improvement from 2.05 on the day of keratoplasty to 1.66 in the latest examination.

### 3.1. Preoperative VA Outcomes for Each Etiology

Fifty-one patients were included in this analysis. There were no statistically significant differences in preoperative VA depending on the etiology of corneal blindness (*p* = 0.68).

### 3.2. Postoperative VA Outcomes for Each Etiology

Forty-nine patients were included in this analysis. There were no statistically significant differences in postoperative VA outcomes depending on etiology (*p* = 0.19).

### 3.3. Differences between Pre- and Postoperative Visual Outcomes for Each Indication for Graft

Fifty patients were included in this analysis. Thirty-two patients noted an improvement in VA. In 18 cases, VA remained unchanged or became worse. There was no statistically significant relationship between VA improvement and etiology (*p* = 0.12) (Figure 3).

### 3.4. Improvement in VA after Keratoplasty

Forty-eight patients were included in this analysis. There was a statistically significant improvement in VA in the latest follow-up (*p* = 0.003) (Figure 4).

### 3.5. Improvement in VA for Each Indication for Graft

The improvement in VA was statistically significant only in the case of infection (*p* = 0.001) (Table 2). A summary of visual outcomes can be found in Table 3.

### 3.6. Infections

Infection was the main indication for keratoplasty in our study group. Bacterial and fungal infections were the most common. The full characteristics of the study group can be found in Table 4.

#### 3.6.1. Preoperative Visual Outcome and the Type of Infection

Twenty-five patients were included in this analysis. There were no statistically significant differences in preoperative visual outcome and the type of infection (*p* = 0.59).

#### 3.6.2. Postoperative Visual Outcome for Different Types of Infection

Twenty-five patients were included in this analysis. There was no statistically significant difference in the postoperative VA depending on the type of infection (*p* = 0.87).

#### 3.6.3. Improvement in Visual Outcome Depending on the Type of Infection

Twenty-four patients were included in this analysis. Eighteen patients noted an improvement in visual outcome. In six cases, visual outcomes remained unchanged or worsened. There was no statistically significant relationship between VA improvement and the type of infection (*p* = 0.5) (Figure 5).

#### 3.6.4. Comparison of VA Improvement for Each Type of Infection

The improvement in VA was statistically significant only in the case of bacterial infection (*p* = 0.047) (Table 5). A summary of all visual outcomes can be found in Table 6.

### 3.7. Graft Survival

The graft survival time was defined as: (a) the time between the first and second keratoplasty; (b) the time between keratoplasty and the discovery of an atrophic eyeball in the postoperative follow-up; and (c) the time between keratoplasty and enucleation of the eyeball. In the uncomplicated cases, the graft survival time was defined as the observation time.

#### 3.7.1. Graft survival Probability

An analysis of the graft survival showed a 1-year survival of 54% and a 5-year survival of 19.8% of grafts (Figure 6).

#### 3.7.2. Graft Survival Probability: Influence of Primary Etiology

There was no statistically significant difference in graft survival for each etiology (*p* = 0.8) (Figure 7).

#### 3.7.3. Graft Survival Probability: Influence of Sex

There was no statistically significant difference in graft survival between females and males (*p* = 0.7) (Figure 8).

#### 3.7.4. Graft Survival Probability: Influence of Previous Surgical Procedures on the Eye

A history of previous surgical procedures on the eye did not have an impact on graft survival in our study group (*p* = 0.5) (Figure 9).

#### 3.7.5. Graft Survival Probability: Influence of the Recipient’s Age

There was no statistically significant difference in graft survival probability depending on the recipient’s age (*p* = 0.7) (Figure 10).

#### 3.7.6. Graft Survival Probability: Influence of Additional Systemic Immunosuppression

There was no statistically significant difference in graft survival for the group with systemic immunosuppressive therapy (*p* = 0.3) (Figure 11).

#### 3.7.7. Graft Survival Probability: Influence of Graft Diameter

Graft diameter did not have an impact on graft survival (*p* = 0.1) (Figure 12).

#### 3.7.8. Graft Survival Probability: Influence of the Number of Keratoplasties

The 1-year survival was 52.8% for the first keratoplasty and 70% for the second keratoplasty. The 5-year survival was 19.9% for the first keratoplasty. None of the second keratoplasties survived 5 years (Figure 13). Previous graft disease (earlier considered as the primary high-risk etiology) was regarded as the second keratoplasty in this analysis. There was no statistically significant difference in graft survival probability for the first and second keratoplasty (*p* = 0.4).

#### 3.7.9. Comparison of Graft Survival Time for Each Etiology

In this analysis, we included 28 patients that required repeat keratoplasty. There were no statistically significant differences in graft survival time depending on etiology (*p* = 0.44).

#### 3.7.10. Graft Survival Probability: Impact of Keratoplasty Combined with Vitrectomy

Within the whole study group, six patients were subjected to keratoplasty combined with vitrectomy. In four of them, vitrectomy was performed together with the first keratoplasty. The primary indications for keratoplasty in this group were trauma (3/4) and infection (1/4). Only one of the patients who had the vitrectomy combined with the first keratoplasty required a second keratoplasty (2 months after the first keratoplasty). The two other patients had a vitrectomy combined with the second keratoplasty. Their primary indications for keratoplasty were infection (1/2) and burns (1/2). The mean observation time in the group without complications was 13 months. Generally, in four patients, VA improved, in one case it remained unchanged, and in another, it deteriorated. The mean VA increased from 2.42 before to 2.08 after the operation, but this change was not statistically significant (*p* = 0.17). Despite quite a long observation time without rejection episodes, functional success was not obtained.

#### 3.7.11. Comparison of Graft Survival Time for Different Types of Infection

In this analysis, we included 14 patients with infectious etiology that required repeat keratoplasty. There were no statistically significant differences in graft survival time depending on the type of infection (*p* = 0.36). Graft survival probability was not statistically significant (*p* = 0.06) (Figure 14).

### 3.8. Summary of the Results

VA in our study group significantly improved from 2.05 logMAR on the day of keratoplasty to 1.66 logMAR in the latest examination (*p* = 0.003). It proves that keratoplasty is an essential method of treatment in improving vision. An analysis of visual improvement performed separately for each etiology showed a statistically significant increase in VA only in the case of infections (*p* = 0.001). It showed poorer outcomes in patients with other etiologies, namely burns, trauma, sterile perforation, and previous graft disease. Further analysis proved patients with bacterial infections to have the best visual outcomes (*p* = 0.047). An analysis of the graft survival showed a 1-year survival of 54% and a 5-year survival of 19.8% of grafts. It proves that high-risk corneal transplants carry a greater risk of rejection than non-high-risk procedures.

## 4. Discussion

The immune privilege of corneal allografts is based on three fundamental processes: afferent blockade, the deviation of the immune response, and the elimination of immune effector elements. The afferent blockade of the immune response is caused by graft bed avascularity [7]. The absence of lymph vessels is also an important factor that leads to the immune privilege of the cornea [8]. Anterior chamber-associated immune deviation (ACAID) generates regulatory T cells (Tregs), which are responsible for the development and maintenance of ocular immune privilege [7,9]. Corneal allografts placed over the anterior chamber are in direct contact with anti-inflammatory and immunosuppressive cytokines from the aqueous humor [7]. In the Collaborative Corneal Transplantation Study, high risk was defined as a cornea with two or more vascularized quadrants or one in which a graft had previously been rejected [10]. High-risk corneal transplantations have lower success rates because of a higher incidence of immune-mediated graft rejection. Factors contributing to a higher risk of immunological rejection are: inflammatory, allergic, or infectious causes of corneal opacity; re-transplantation; corneal neovascularization and neo-lymphangiogenesis; glaucoma history; prior ocular surgery; blood transfusion history; larger donor cornea size; surgical complications; lens status; and male-to-female transplantation [3]. The presence of blood and lymph vessels reduces the immune privilege of corneal allografts by promoting the migration, recruitment, and infiltration of immune effector elements [8]. Immune-mediated graft rejection occurs when the main layers of the cornea are destroyed by the immune response. The rejection of the endothelium leads to irreversible endothelial cell loss and may result in permanent graft failure [11].

The probability of graft survival after penetrating keratoplasty was 86% at 1 year, 73% at 5 years, and 55% at 15 years [12]. Low-risk penetrating keratoplasty represents an even better survival rate of 90% at 5 years and 82% at 10 years [13]. High-risk recipients have a lower survival rate with a 5-year survival of 54.2% compared to 91.3% for primarily non-inflamed eyes [14]. A total of 70% of the high-risk grafts may face failure due to corneal bed vascularization or previous graft rejection [11,15]. In high-risk corneal recipients, rejection episodes occur in 40–70% of cases a year [11,16]. In our study, the 1-year survival probability for allografts was 54%, which is comparable to other studies. An increased risk of rejection was seen in young recipients [17]. However, our study did not confirm this relationship.

Repeat transplantation represented poorer outcomes [18,19,20]. The 5-year survival rate decreased from 72.5% in the first to 37.3–53.4% in repeat transplants [19]. Failure rates in repeat transplants were 17% at 2 years compared to 6% in first transplants [18]. In our study, the first and second keratoplasty involved a 1-year survival rate of 52.8% and 70%, respectively. This difference, however, was not statistically significant. The outcomes might have been influenced by the fact that we compared high-risk indications with repeat keratoplasty, which itself is a high-risk indication. It may suggest that all high-risk corneal transplantations have similar survival rates.

The Australian Corneal Graft Registry Report (ACGR) showed a poorer survival of grafts with a diameter of less than 7.75 mm and more than 8.5 mm [20]. Our study did not show statistically significant differences in graft survival for different graft diameters.

Some studies prove penetrating keratoplasty combined with vitrectomy to be a safe and effective procedure [21,22,23,24,25]. However, another study showed poorer outcomes in patients with severe ocular trauma [26]. In our study, only one out of six patients required repeat keratoplasty after this procedure, but functional success was not obtained.

Topical corticosteroids (CS) (mainly prednisolone and dexamethasone) are routinely used for the prevention and treatment of corneal graft rejection [27]. Difluprednate may be effective in penetrating keratoplasty graft rejection treatment, especially in non-high-risk grafts [28]. High-risk corneal allografts require more intensive treatment with gradually reduced doses of CS over a period of 6–12 months [17]. Besides the most common topical route, CS may also be administered by subconjunctival or systemic route [29]. However, the use of CS in corneal graft rejection treatment is limited by their ocular side effects, including infections, impaired wound healing, cataract formation, and glaucoma, as well as systemic side effects [11,17,30,31].

Cyclosporine is a calcineurin inhibitor used as an immunosuppressive agent in solid organ transplants. Topical cyclosporine was used in 48% of high-risk grafts [29]. Topical cyclosporine A (CsA) was not as effective as topical prednisolone in the prevention of graft rejection [32]. However, in another study, a combined regimen of topical CS and topical CsA was associated with a higher 1- and 2-year rejection-free graft survival rate [33]. The results of systemic CsA in the prevention of high-risk corneal transplantation rejection were inconsistent [17]. The most common adverse events of CsA therapy in high-risk corneal transplants were herpes keratitis and hypertension [34].

MMF is another systemic immunosuppressive agent used in high-risk corneal transplants. Reinhard et al. established a 1-year immune reaction-free rate of 89% in the MMF group, as compared to only 67% in the control group [35].

Other pharmacotherapeutic agents that can be used in high-risk corneal transplantation are tacrolimus and rapamycin [17].

Every patient from our study group received topical CS. Some of them were given additional systemic CS, MMF, or systemic CS + MMF. Our study showed no statistically significant difference in graft survival depending on the postoperative prophylaxis regimen.

We proved that keratoplasty was necessary and successful in restoring our patients’ vision, as our study group noted a statistically significant VA improvement following keratoplasty. Considering functional success for each etiology separately, we revealed that only patients with infectious indications for keratoplasty had statistically significant VA improvement. This indicates that the management of keratoplasties after traumas, burns, previous graft diseases, and sterile perforations should be reconsidered to improve their outcomes in the future. Of all patients with an infectious etiology of corneal blindness, those with bacterial infections noted the best functional results. It is, therefore, necessary to determine the cause of worse outcomes among patients with viral, fungal, and amebic infections so as to improve their treatment.

Our study has some limitations. First of all, a greater sample size and longer observation time are required, especially in the case of less common indications for corneal transplantation, such as burns, sterile perforations, or Acanthamoeba keratitis. It is also impossible to estimate the significance of each factor accurately, as many factors can impact graft survival.

However, our study showed that similar outcomes can be reached despite the different indications for corneal transplantation. Many patients noted an improvement in VA, which proves that this treatment is essential in restoring visual functions. We proved the best functional success for bacterial infections among the infectious indications for keratoplasty. Our study showed that systemic immunosuppression with CS or MMF is necessary in many cases.

## 5. Conclusions

High-risk corneal transplants have poorer outcomes than routinely performed corneal transplants. Despite the lower survival rate, our study group noted statistically significant improvement in VA, which was equivalent to functional success. More studies should be performed for a better understanding of high-risk settings. It is important to establish the factors that may impact graft survival to provide better routine management that would increase the survival rate in the future.

## Figures and Tables

**Figure 1 jcm-11-05511-f001:**
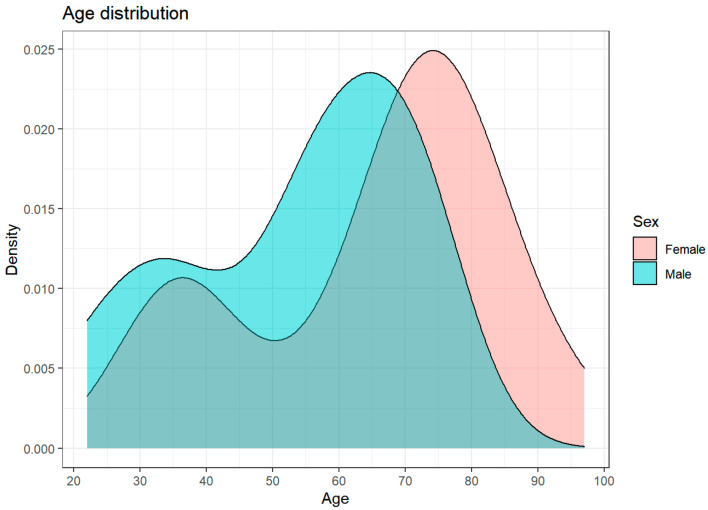
Age distribution with two peaks caused by traumas (first peak) and infections (second peak), being the main indications for keratoplasty in younger and older patients, respectively.

**Figure 2 jcm-11-05511-f002:**
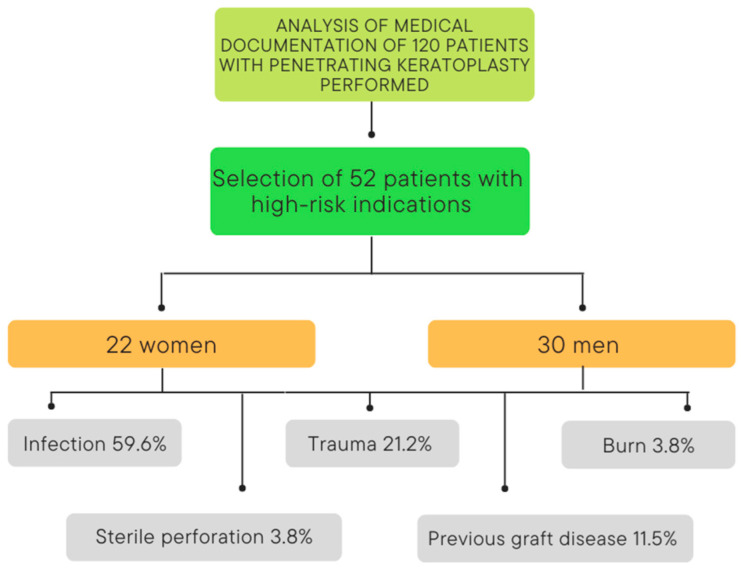
Structure of the study group, according to sex and the etiology of corneal blindness.

**Figure 3 jcm-11-05511-f003:**
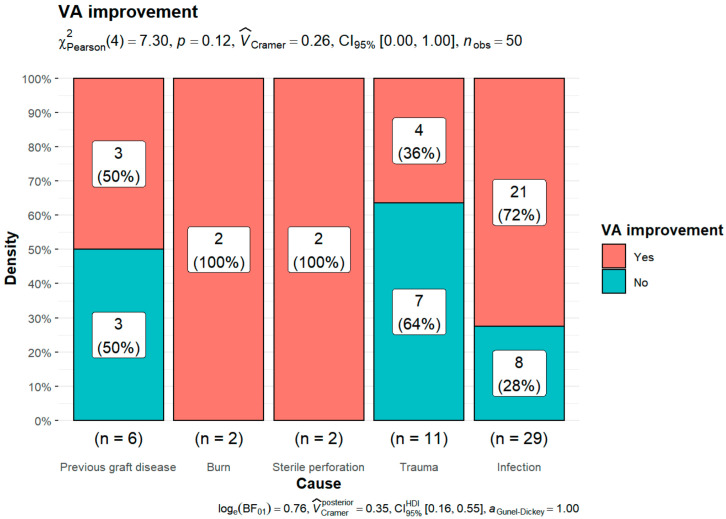
Relationship between VA improvement and etiology of corneal blindness.

**Figure 4 jcm-11-05511-f004:**
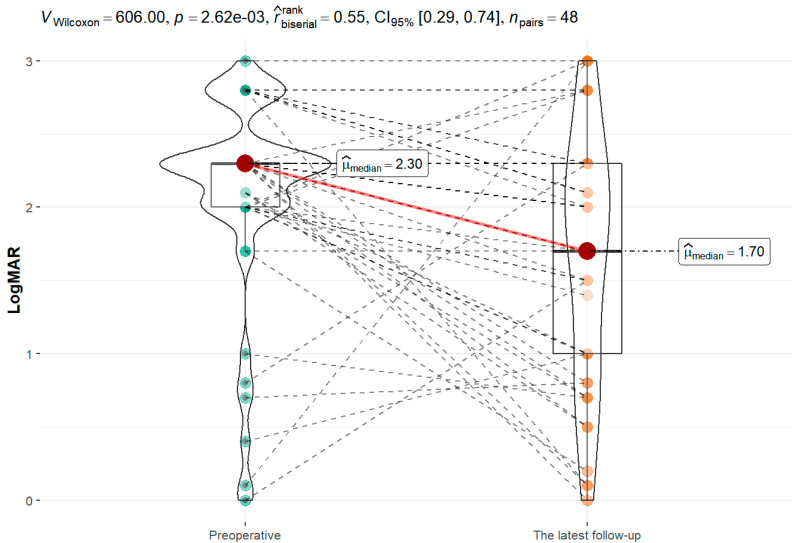
Visual outcomes before and after keratoplasty.

**Figure 5 jcm-11-05511-f005:**
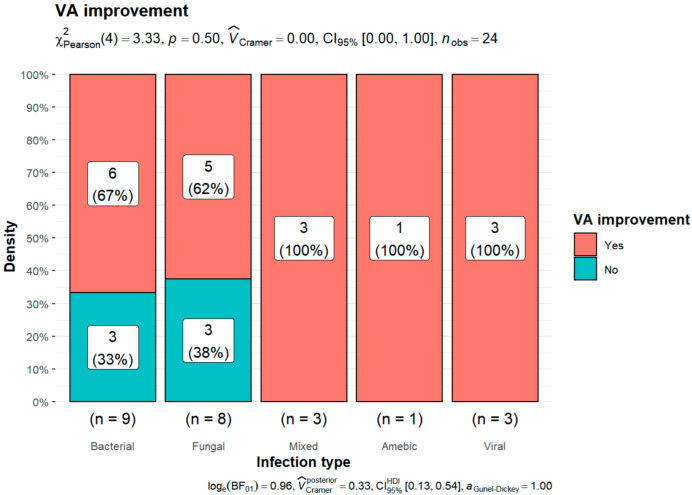
Improvement in VA and the type of infection.

**Figure 6 jcm-11-05511-f006:**
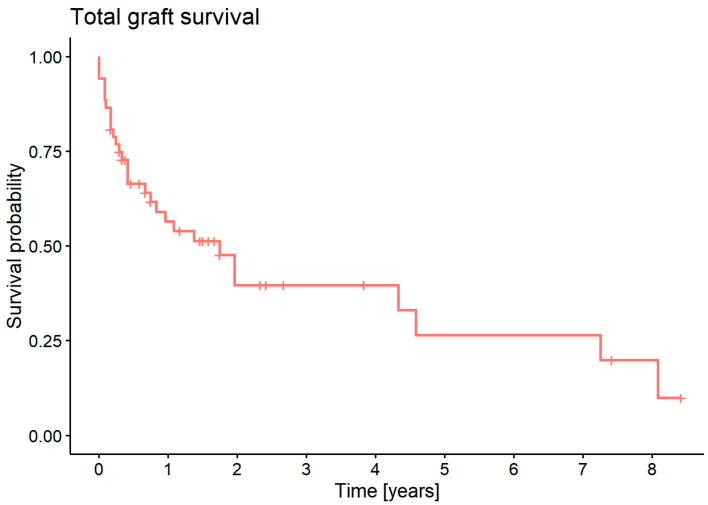
Overall graft survival probability.

**Figure 7 jcm-11-05511-f007:**
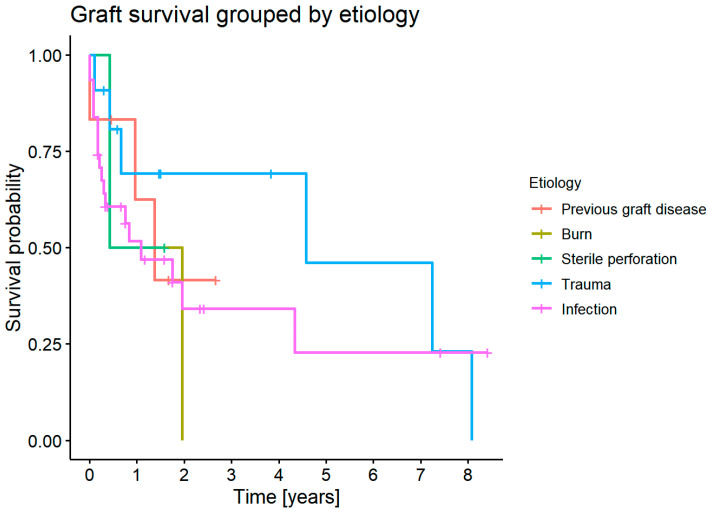
Survival probability for each etiology.

**Figure 8 jcm-11-05511-f008:**
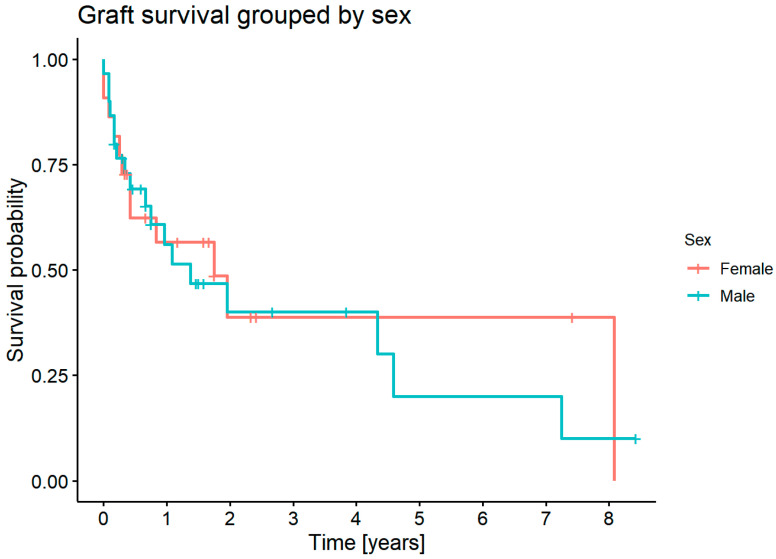
Survival probability for females and males.

**Figure 9 jcm-11-05511-f009:**
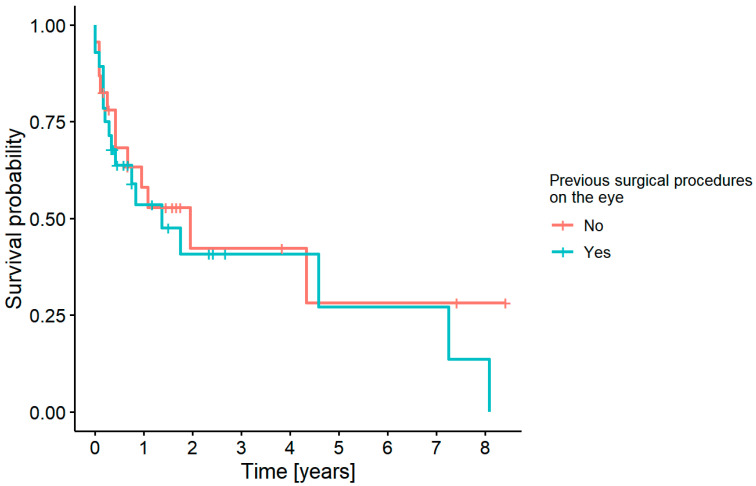
Survival probability and the history of previous surgical procedures.

**Figure 10 jcm-11-05511-f010:**
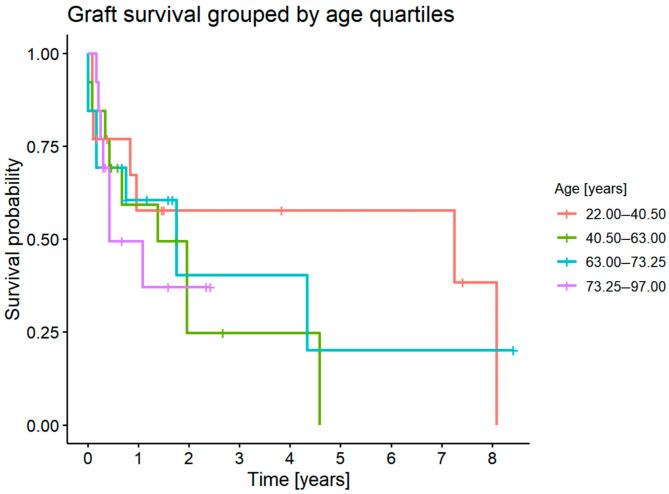
Graft survival grouped by age quartiles.

**Figure 11 jcm-11-05511-f011:**
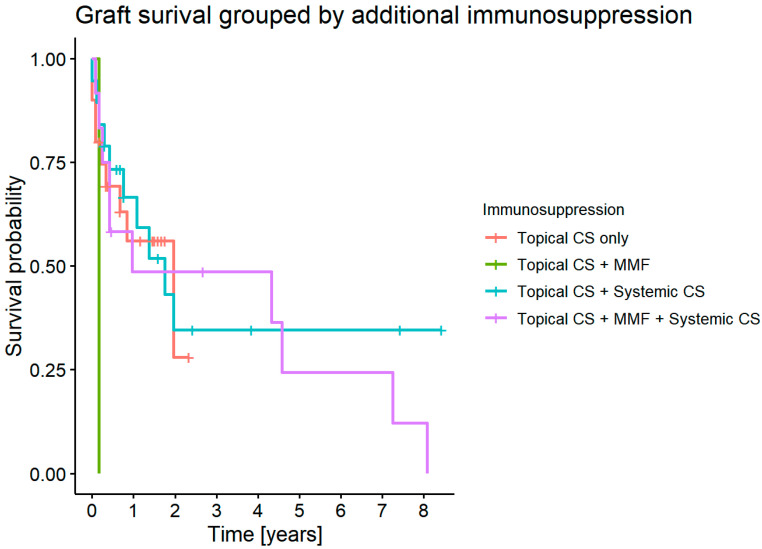
Graft survival grouped by the use of immunosuppression.

**Figure 12 jcm-11-05511-f012:**
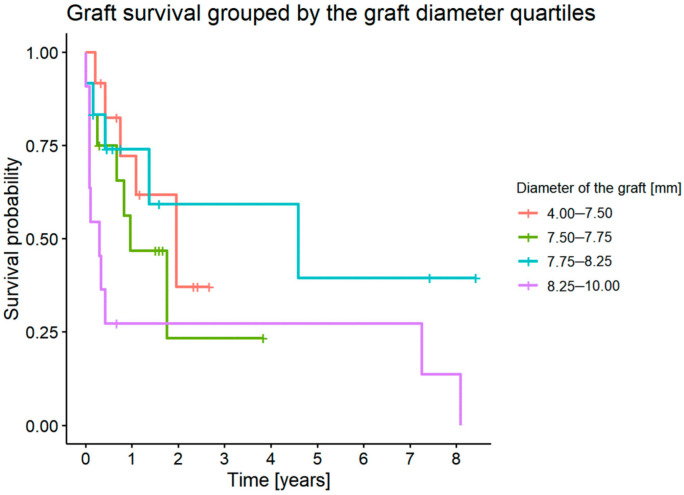
Graft survival grouped by the first graft diameter.

**Figure 13 jcm-11-05511-f013:**
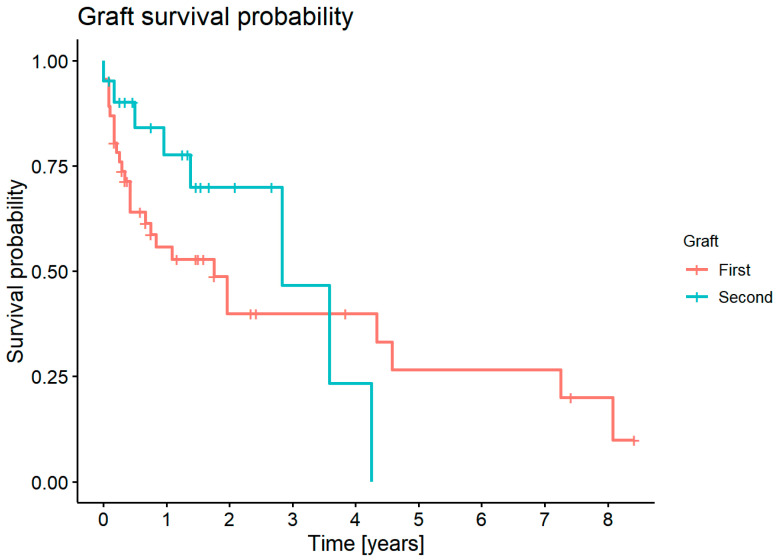
Graft survival probability for the first and second keratoplasty.

**Figure 14 jcm-11-05511-f014:**
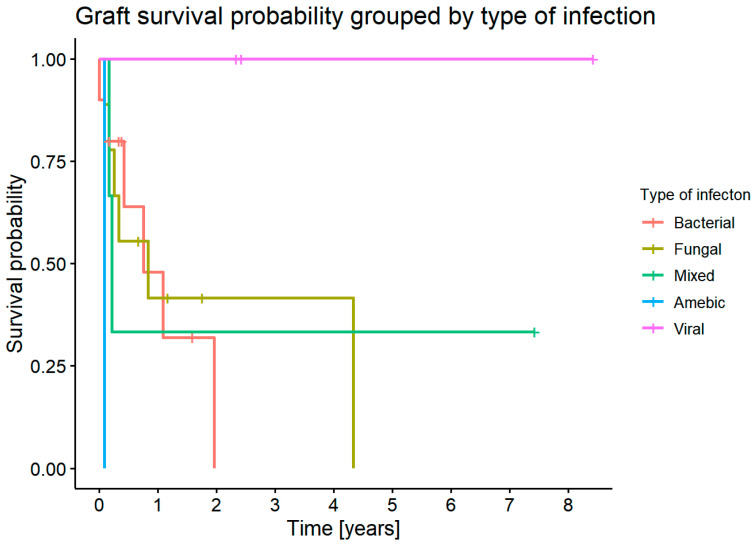
Graft survival probability for each type of infection.

**Table 1 jcm-11-05511-t001:** Characteristic of the study group.

	Infection	Trauma	Previous Graft Disease	Burn	Sterile Perforation
	N = 31	N = 11	N = 6	N = 2	N = 2
	59.6%	21.2%	11.5%	3.8%	3.8%
Female	16 (51.6%)	1 (9.1%)	2 (33.3%)	1 (50.0%)	2 (100.0%)
Male	15 (48.4%)	10 (90.9%)	4 (66,7%)	1 (50.0%)	0 (0.0%)
Mean age (SD) (years)	63.4 (16.8)	42.6 (18.8)	57.3 (12.9)	57.5 (23.3)	80.0 (7.1)
Range	34.0–97.0	22.0–74.0	33.0–70.0	41.0–74.0	75.0–85.0

**Table 2 jcm-11-05511-t002:** Improvement in VA and etiology.

Indication for Graft	N	*p*-Value
Previous graft disease	6	0.387
Burn	2	0.333
Sterile perforation	2	0.167
Trauma	11	0.605
Infection	31	**0.001**

**Table 3 jcm-11-05511-t003:** Summary of the visual outcomes for each indication for keratoplasty.

	Previous Graft Disease	Burn	Sterile Perforation	Trauma	Infection
	(N = 6)	(N = 2)	(N = 2)	(N = 11)	(N = 31)
	11.5%	3.8%	3.8%	21.2%	59.6%
**Preoperative logMAR**					
Mean (SD)	1.69 (0.96)	2.40 (0.57)	2.55 (0.35)	1.89 (0.98)	2.12 (0.59)
1st Quartile	1.18	2.20	2.42	1.85	2.00
Median	2.30	2.40	2.55	2.30	2.30
3rd Quartile	2.30	2.60	2.67	2.30	2.30
Min–Max	0.16–2.30	2.00–2.80	2.30–2.80	0.00–3.00	0.40–2.80
**The last follow-up logMAR**					
Mean (SD)	1.46 (1.18)	2.00 (0.42)	1.05 (1.34)	2.19 (0.56)	1.52 (0.80)
1st Quartile	0.50	1.85	0.58	1.85	0.80
Median	1.70	2.00	01.05	2.10	1.50
3rd Quartile	2.30	2.15	1.52	2.55	2.10
Min–Max	0.00–2.80	1.70–2.30	0.10–2.00	1.40–3.00	0.10–3.00

**Table 4 jcm-11-05511-t004:** Characteristics of the study group with infections.

	Bacterial	Fungal	Mixed	Amebic	Viral
	(N = 10)	(N = 9)	(N = 3)	(N = 1)	(N = 3)
	38.5%	34.6%	11.5%	3.8%	11.5%
Female	5 (50.0%)	6 (66.7%)	2 (66.7%)	0 (0.0%)	2 (66.7%)
Male	5 (50.0%)	3 (33.3%)	1 (33.35)	1 (100.0%)	1 (33.3%)
Mean age (SD) (years)	60.1 (16.7)	63.0 (21.2)	63.0 (21.7)	49.0 (NA)	74.7 (6.5)
Range	35.0–75.0	34.0–97.0	38.0–77.0	49.0–49.0	68.0–81.0

**Table 5 jcm-11-05511-t005:** Improvement in VA for each type of infection.

Infection Type	Quantity	*p*-Value (Student’s *t*-Test)	*p*-Value (Wilcoxon Test)
Bacterial	9	-	**0.047**
Fungal	9	-	0.072
Mixed	3	0.152	-
Amebic	1	-	0.500
Viral	3	0.109	-

**Table 6 jcm-11-05511-t006:** Summary of visual outcomes for each type of infection.

	Bacterial	Fungal	Mixed	Viral	Amebic
	(N = 10)	(N = 9)	(N = 3)	(N = 3)	(N = 1)
	38.5%	34.6%	11.5%	11.5%	3.8%
**Preoperative logMAR**					
Mean (SD)	2.11 (0.63)	2.11 (0.76)	2.27 (0.55)	1.70 (0.61)	2.30 (NA)
1st Quartile	2.10	2.00	2.00	1.50	2.30
Median	2.30	2.30	2.30	2.00	2.30
3rd Quartile	2.30	2.42	2.55	02.05	2.30
Min–Max	0.70–2.80	0.40–2.80	1.70–2.80	1.00–2.10	2.30–2.30
**The last follow-up logMAR**					
Mean (SD)	1.42 (0.89)	1.53 (0.73)	1.40 (1.08)	1.07 (0.40)	2.00 (NA)
1st Quartile	0.70	1.00	0.95	0.85	2.00
Median	1.70	1.50	1.70	1.00	2.00
3rd Quartile	2.30	2.10	2.00	1.25	2.00
Min–Max	0.10–2.30	0.70–2.80	0.20–2.30	0.70–1.50	2.00–2.00

## Data Availability

Not applicable.

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
