# Peer review of "Management and Treatment Outcomes of High-Risk Corneal Transplantations"

_jcm, 2022, doi:10.3390/jcm11195511_

Round 1

Reviewer 1 Report

Major comments:

1. Sections 3.7.9, 3.7.10 and 3.7.11 - more detailed explanation is needed to describe the criteria of patients inclusion in the analysis: 28, 6 and 14 patients respectively.

2. Table 4. More information in methods section is needed to describe the method of infection etiology confirmation: what analysis were used to differentiate viral, bacterial or fungal infection?

3. Figures 9 - 16 more information on how survival probability was calculated is needed. 

4. Figures 2,3,6,7,17 and 18 - I would recommend to reduce the number of pictures of the same type, because they do not increase the descriptiveness of the data. 

Minor comments:

1. AC - aqueous humor - abbreviation is just listed, but now used in the article

2. Table 2. 9 digits after point - such detalisation of p-value is unnecessary (i.e. p=0.386968823), it doesn't add scientific and clinical sense and accuracy, it just confuses the reader. Three points is more than enough: p=0.387

Author Response

Response to major comments

  1. We added the explanations in section 3.7.9, 3.7.10 and 3.7.11
  2. We added he required information to the methods section
  3. We added the required information to the methods section
  4. We reduced the number of the figures

Response to minor comments:

  1. We removed this abbreviation
  2. Changed according to the reviewer’s suggestion

Reviewer 2 Report

Thank you to the authors for this work. It is a well written manuscript on the whole, but the introduction can be improved in terms of syntax and flow.

This paper provides a good retrospective review of hot grafts and outcomes. This will be of interest to readers worldwide. 

Author Response

We rewrote the introduction focusing on the aspects of syntax and flow.

Reviewer 3 Report

Reviewer

General comments

This paper is an accurately, needed, and useful describing management and treatment Outcomes of high-risk corneal transplantations.

This research is relatively complete, but there are still some comments that have to be addressed.

Introduction

In addition to mentioning the main objective it should define the purpose of the work and its significance and highlight the main conclusions.

Materials and Methods

Add a flow diagram to population and their distribution.

Sample size calculation details need to be provided.

Results

The results section is hard to read.

Must provide a summary description of the experimental results and their interpretation.

Discussion

Authors should discuss the results and their implications.

Finally, what is the key message from this study?

What is the practical impact of this study?

Author Response

Introduction

We added the appropriate information at the end of the introduction.

Materials and methods

We added the flow diagram (figure 2)

At the end of the materials and methods section we added the information about the the size of the sample that was chosen for each analysis. We added the information about the number of patients whose documentation was analyzed (120 patients).

Results

We added a new section 3.8 containing a summary of our results

Discussion

We added a summary of implications, key message and practical impact of our results to the discussion.